# Skill retention with ultrasound curricula

**Lawrence Matthews[1], Krysta Contino[1], Charlotte Nussbaum[1], Krystal Hunter[1,2], Christa Schorr[1,2], Nitin Puri[1,2]***

**1** Cooper University Health Care, Camden, NJ, United States of America, **2** Cooper Medical School of Rowan University, Camden, NJ, United States of America

* Puri-nitin@cooperhealth.edu

## Abstract

### Background

Implementation of a point of care ultrasound curricula is valuable, but optimal integration for internal medicine residency is unclear. The purpose of this study was to evaluate if a structured ultrasound curriculum vs. structured ultrasound curriculum plus supervised thoracic ultrasounds would improve internal medicine residents' skill and retention 6 and 12 months from baseline.

### Methods

We conducted a randomized controlled study evaluating internal medical residents' skill retention of thoracic ultrasound using a structured curriculum (control, n = 14) vs. structured curriculum plus 20 supervised bedside thoracic ultrasounds (intervention, n = 14). We used a stratified randomization based on program year. All subjects attended a half-day course that included 5 lectures and hands-on sessions at baseline. Assessments included written and practical exams at baseline, immediately post-course and at 6 and 12 months. Scores are reported as a percentage for the number of correct responses/number of questions (range 0–100%). The Mann Whitney U and the Friedman tests were used for analyses.

### Results

Twenty-eight residents were enrolled. Two subjects withdrew prior to the 6-month exams. Written exam scores for all subjects improved, baseline median (IQR) 60 (46.47 to 66.67) post-course 80 (65 to 86.67), 6-month 80 (66.67 to 86.67) and 12-month 86.67 (80 to 88.34), p = <0.001. All subjects practical exam scores median (IQR) significantly improved, baseline 18.18 (7.95 to 32.95), post-course 59.09 (45.45 to 70.45), 6 month 71.74 (60.87 to 82.61) and 12-month 76.09 (65.22 to 88.05), p = <0.001. Comparing the control group to the intervention group, there were statistically significant higher scores, median (IQR), in the intervention group on the practical exam at 6 months 63.05 (48.92 to 69.57) vs. 82.61(72.83 to89.13), p = <0.001.

### Conclusion

In this cohort, internal medicine residents participating in a structured thoracic ultrasound course plus 20-supervised ultrasounds achieved higher practical exam scores long-term compared to controls.

**Data Availability Statement:** All relevant data are within the manuscript and its Supporting Information files.

**Funding:** This project was supported by an educational grant from the Cooper Research Institute.

**Competing interests:** The authors have declared that no competing interests exist.

## Introduction

Educators agree on the value of implementation of point of care ultrasound into Internal Medicine curricula is valuable, but optimal integration to sustains skills in point of care ultrasound (POCUS) curricula into an Internal Medicine Residency is unclear [1, 2]. The utility of an ultrasound workshop combined with a designated curriculum has proven beneficial in minimizing attrition rate, but did not address the practical acquisition and interpretation of images [3, 4]. Longitudinal ultrasound curricula has been shown to increase knowledge retention and improve skill acquisition in internal medicine (IM) residents, but skill acquisition retention has not been fully quantified [5]. IM physicians were able to accurately assess left systolic function using focused cardiac ultrasound with a rigorous training program in one study [6]. The purpose of this study was to evaluate thoracic ultrasound skill retention comparing internal medicine (IM) residents completing a standard ultrasound curriculum versus a standard curriculum plus obtaining 20 supervised bedside thoracic ultrasounds.

## Materials and methods

We conducted a randomized controlled study from June 2017 to June 2018. All post-graduate year (PGY) -1 and PGY-2 residents attending the ultrasound curriculum were invited to participate. Categorical PGY-1 and PGY-2 IM residents who attended the one-day ultrasound training course and signed the informed consent form were included. Participants were randomized to either the standard ultrasound curriculum (control group) or standard ultrasound curriculum plus obtaining 20 supervised thoracic ultrasounds at the bedside (intervention group). We used a stratified randomization based on program year. To determine group allocation, subjects were assigned using Excel random numbers.

All residents participated in the standard internal medicine residency ultrasound curriculum that included:

1. A half-day comprehensive course at the beginning of the study comprised of vascular, cardiac and thoracic ultrasound lectures. All lectures were one hour. Pathology image review and hands-on scanning of models (3:1 student to faculty ratio), included one hour of scanning for thoracic ultrasound. All course instructors, for the comprehensive course and teaching sessions completed throughout the academic year, were either internal medicine faculty, critical care fellows, critical care attending physicians or ultrasound fellowship trained emergency medicine physicians.

2. Five one-hour lectures, on physics, abdominal, vascular, cardiac, and thoracic ultrasound spanned throughout the academic year, and

3. Five concordant (within the same week as the lecture) one hour hands-on sessions, based on the lecture topics, using standardized patients at our Simulation Center.

The intervention group was required to complete 20 additional supervised ultrasound scans of a hemi-thorax on hospitalized patients admitted to the general medical floors or the medical-surgical intensive care unit (ICU). Participants completed 10 scans between baseline and the 6-month assessment and 10 additional scans between 6 months and the 12-month assessment. Three study investigators (NT, LM, CN) proficient in thoracic ultrasound supervised the scans. All the essential parts of a comprehensive thoracic ultrasound were reviewed with the subject in the intervention group by one of the three study investigators (NT, LM, CN) and feedback was provided to the participating resident in real-time. Proficiency of the instructors was determined by following the Canadian recommendations for critical care ultrasound training and competency in thoracic ultrasound [7].

The control group participants could perform scans per their own discretion. Data was not captured for non-study related scans. One of three ultrasound machines (Sonosite M-Turbo, Bothell, WA, Sonosite X-porte, Bothell, WA, Mindray North America Mahwah, NJ) were used, depending on machine availability.

Assessment for both the intervention and control groups occurred at four intervals: immediately before the half-day ultrasound course at the beginning of the study, immediately after the half-day course, at 6 months, and at 12 months. Assessment consisted of both a written test and a practical test designed by the study investigators (NP, LM, KC, CN).

In this study, competency with POCUS is consistent with a conceptual framework that includes assessment of the acquisition and application of POCUS related knowledge, demonstration of technical skills, and effective integration into clinical practice [8]. The written and practical evaluations were developed based on the International Thoracic Ultrasound Guidelines and the American College of Chest Physicians Guidelines [9–11]. Prior to the study start, the assessment was reviewed and edited by 10 volunteers, including a novice (< 1 year experience), those proficient in thoracic ultrasound (> 3 years experience) and clinicians advanced in thoracic ultrasound (regional and national faculty for POCUS). The written assessment included 15 multiple-choice questions. Eight questions sought a diagnosis based on a projected still image or short video clip, five questions sought management understanding based on a projected still image or clip, and two questions were assessing general ultrasound knowledge without corresponding images. (S1 Fig) The practical assessment was five minutes in length and required the participant to demonstrate 23 different competencies in thoracic ultrasound imaging. (Table 1) We chose these ultrasound skills, as they are clinically applicable for internists [9].

We completed an analysis over time, examining the baseline, immediate posttest, at 6 months and 12 months. In order to determine which statistical test to run, we ran a Shapiro Wilke test and test of skewness, which indicated that some of these data were not normally distributed. In an effort to make the statistical testing consistent, we ran all repeated measures non-parametric testing using the Friedman test. An analysis of separate test scores performed via the Mann Whitney U test. Results are presented as median (interquartile range, IQR). Scores are reported as a percentage of the number of correct responses/number of questions (range 0–100%). We completed a Bonferroni correction in order to account for the 14 tests that were run. This created a significant level of $p < = 0.004$. The Cooper Institutional Review Board reviewed and approved the study.

## Results

Thirty-six residents were invited to participate. Fourteen PGY-1 and fourteen PGY-2 residents were enrolled in May 2017. The intervention group included 11 male and 3 female residents, six PGY-1, and eight PGY-2. The control group included 7 male and 7 female residents, eight PGY-1 and six PGY-2. Twenty-six of the 28 participants completed the study. In the control group, one of the PGY-2 residents voluntarily withdrew consent from the study and another was unable to complete the 6- and 12-month follow-up. All of the intervention group participants completed the required supervised scans at 6 and 12 months.

There was a significant difference in written and practical exam scores over time in both the intervention ($n = 14$, $p = < 0.001$) and control ($n = 12$, $p = <0.001$ groups from baseline to 12 months. (Fig 1) Written exam scores from baseline to 12 months, increased by approximately 26.67 points and the practical exam scores increased by approximately 53.53 points.

When comparing the intervention group to the control group, no significant differences were observed for any of the four written assessments. (Table 2) Counts of correct responses

**Table 1. Thoracic ultrasound competencies.**

| Image acquisition and identification of anatomy |
| --- |
| 1.    Identify muscle wall |
| 2.    Identify pleural line |
| 3.    Identify pleural sliding |
| 4.    Identify rib shadows |
| 5.    Identify muscle wall in M mode |
| 6.    Identify pleural line in M mode |
| 7.    Identify pleural sliding in M mode |
| 8.    Identify a-lines |
| 9.    Identify b-lines if present |
| 10.  Identify z-lines |
| 11.  Identify lung pulse if present |
| 12.  Identify presence or absence of lung point |
| 13.  Identify diaphragm |
| 14.  Identify liver |
| 15.  Identify spleen |
| 16.  Identify caudad vs cephalad |
| **Examine enough sites to rule out pneumothorax and pleural effusion on each side of the thorax** |
| 17.  Evaluate 4 sites on each side for lung sliding |
| 18.  Evaluate for pleural effusion posteriorly bilaterally |
| **Image Optimization**: Adjust image to assess pleura/lungs |
| 19.  Adjust gain |
| 20.  Adjust depth |
| 21.  Orient probe marker |
| 22.  Operator faces machine |
| 23.  Correct transducer setting is chosen on ultrasound machine |

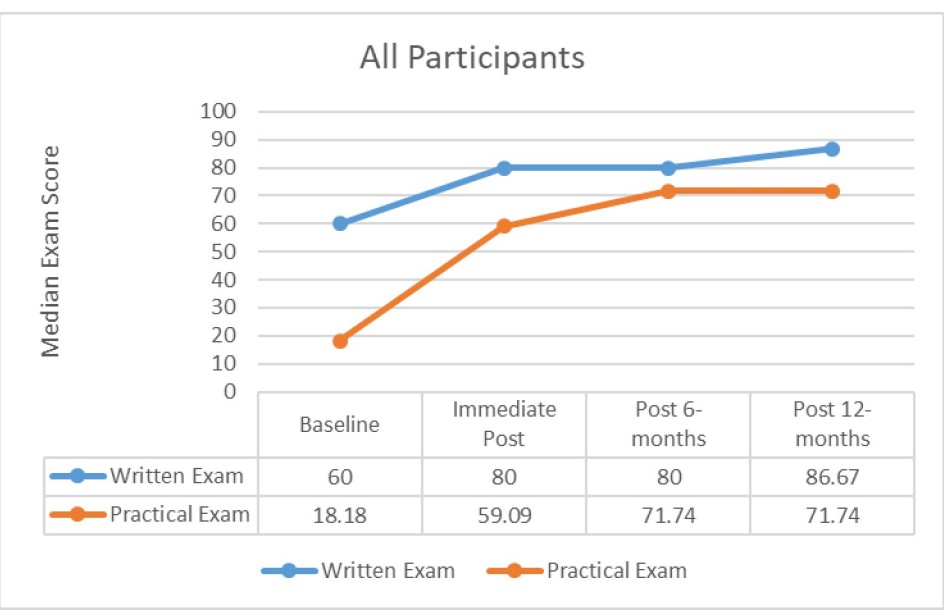

| | Baseline | Immediate Post | Post 6-months | Post 12-months |
| --- | --- | --- | --- | --- |
| Written Exam | 60 | 80 | 80 | 86.67 |
| Practical Exam | 18.18 | 59.09 | 71.74 | 71.74 |

**Fig 1. Exam scores over time for all participants.**

**Table 2. Written exam scores.**

| | Control | | Intervention | | P-value | 95% Confidence Interval |
|---|---|---|---|---|---|---|
| | N | Median (IQR) | N | Median (IQR) | | |
| Pre-course score | 14 | 63.33 (53.33–66.67) | 14 | 53.33 (46.67–68.33) | 0.38 | -13.33 to 6.67 |
| Post-course score | 14 | 80 (60–86.67) | 14 | 76.67 (63.33–81.67) | 0.91 | -13.33 to 13.33 |
| 6 month score | 12 | 73.33 (66.67–80) | 14 | 83.34 (70–86.67) | 0.18 | -6.66 to 20 |
| 12 month score | 12 | 83.34 (80–86.67) | 14 | 86.67 (80–93.33) | 0.30 | 0 to 13.33 |

IQR = Interquartile range

for the individual written exam questions are available in the S1 Table. The baseline and immediately post-course practical assessment scores were similar between the two groups. A significant difference in the median (IQR) practical assessment score was observed between the two groups at 6 months 82.61 (72.83–89.13) intervention vs. 63.05 (48.92–69.57) control, = $p$ = <0.001. (Table 3)

## Discussion

Our competency based medical education study used a learner driven process of direct observations by supervisors to achieve practical goals. This model allowed both the teachers and the students to take responsibility for achieving success. An alternative yearlong lecture based model, even with handheld ultrasound device (HUD), showed no improvement in knowledge or image interpretation score [12].

In our study, both the control and intervention groups were able to perform non-study related scans, during the study period. Non-study related scans were not counted or recorded. The intervention group achieved competency in thoracic ultrasound through investigator-supervised scans as opposed to the control group, which relied on the traditional didactic educational model.

This single center study demonstrated that a longitudinal practical curriculum improves internal medicine residents' long-term image acquisition proficiency. Our results are similar to Town et al. who showed that residents who participated in hands-on assessments demonstrated improvement in their ultrasound skills (performing 2-point compression to assess for deep vein thrombosis, identification of internal jugular vein and inferior vena cava) over the course of a year [13].

The practical assessment scores for the intervention group decreased from the 6-month assessment to the 12-month assessment, indicating that there may have been a decline in knowledge retention. There were no significant differences in the 6 month nor 12 month scores in the assessment (written, p = 0.08; practical, p = 0.123). The 13% drop in the

**Table 3. Practical assessment scores.**

| | Control | | Intervention | | P-value | 95% Confidence Interval |
|---|---|---|---|---|---|---|
| | N | Median (IQR) | N | Median (IQR) | | |
| Pre-course score | 14 | 15.91 (9.09–28.41) | 14 | 20.46 (4.55–36.36) | 0.70 | -9.09 to 13.64 |
| Post-course score | 14 | 59.09 (48.86–78.41) | 14 | 54.55 (39.77–68.18) | 0.33 | -22.72 to 9.09 |
| 6 month score | 12 | 63.05 (48.92–69.57) | 14 | 86.36 (76.14–93.18) | < 0.001 | 13.04 to 34.78 |
| 12 month score | 12 | 65.22 (47.83–81.52) | 14 | 76.09 (65.22–88.05) | 0.12 | -4.35 to 26.08 |

IQR = Interquartile range

intervention group's practical score is less than prior reports of up to 29% of physicians not being able to replicate their POCUS skillset at one-year after graduating an internal medicine residency [14]. This finding is likely due to the intervention group continuing study participation through one year in our study.

We identified several study limitations. We did not record the number of ultrasounds completed by the control group during the study period. It is unknown if the group performed any scans during the study period, which may have affected the study results. The randomization process did not equally distribute the PGY-1 and PGY-2 residents between the control and intervention groups. Yet it is unknown if this uneven distribution affected study results. Study investigators were the supervisors for the required 20 scans performed by the intervention group. Two investigators (LM and NP) supervised the majority (83.9%) of the proctored scans and performed the majority of standardized assessments. The investigators served as the models and scorers for the practical assessment scans completed by the participants in the intervention group. Different ultrasound machines were used for the practical assessment depending on which one of the three was available, potentially resulting in participants having varying degrees of familiarity with the machines. Two participants in the control group did not complete the entire standard ultrasound curriculum due withdraw in one and personal commitments in another.

We provided examination dates in advance, possibly allowing the subjects to study prior to the examination. Yet, we did not deter subjects from studying at any time during the yearlong study. Use of the same written assessment exam at each time point may have contributed to the improvement in test scores. However, the correct answers were not provided to the subjects and all tests were numbered and accounted for to ensure that no exams were missing.

External experts in ultrasound education reviewed the written and practical assessment exams, but multiple choice and practical questions were not validated prior to study start [15, 16]. Additional reliability and validation studies are need to establish a standardized method to evaluate POCUS images [17].

## Conclusions

In this study, an ultrasound curriculum improved internal medicine residents' written knowledge of thoracic point of care ultrasound. The addition of 20 required supervised thoracic ultrasound scans to a standard lecture-based and hands-on ultrasound curriculum improved internal medicine residents' practical thoracic ultrasound assessment scores at 6 months compared to the control group. Proctored scanning is associated with an observed difference in practical skills overtime. Future studies should include validation of assessment tools, structured time points for the additional supervised scans, and standard procedures for designing and scoring practical assessment.

## Supporting information

**S1 Fig. Written exam questions.**
(DOCX)

**S1 Table. Counts of correct responses for the individual written exam questions.**
(DOCX)

## Author Contributions

**Conceptualization:** Krysta Contino, Charlotte Nussbaum, Krystal Hunter.

**Data curation:** Lawrence Matthews, Krysta Contino, Charlotte Nussbaum, Krystal Hunter, Christa Schorr, Nitin Puri.

**Formal analysis:** Lawrence Matthews, Charlotte Nussbaum, Krystal Hunter, Christa Schorr, Nitin Puri.

**Investigation:** Lawrence Matthews, Charlotte Nussbaum, Christa Schorr, Nitin Puri.

**Methodology:** Lawrence Matthews, Charlotte Nussbaum, Krystal Hunter, Christa Schorr, Nitin Puri.

**Project administration:** Lawrence Matthews, Krysta Contino, Charlotte Nussbaum, Nitin Puri.

**Resources:** Nitin Puri.

**Supervision:** Krysta Contino, Charlotte Nussbaum, Nitin Puri.

**Validation:** Krysta Contino, Charlotte Nussbaum, Christa Schorr, Nitin Puri.

**Visualization:** Charlotte Nussbaum, Christa Schorr, Nitin Puri.

**Writing – original draft:** Lawrence Matthews, Krysta Contino, Charlotte Nussbaum, Krystal Hunter, Christa Schorr, Nitin Puri.

**Writing – review & editing:** Lawrence Matthews, Krysta Contino, Charlotte Nussbaum, Krystal Hunter, Christa Schorr, Nitin Puri.

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
