## [Decision Letter · Decision Letter 0]

6 Aug 2020

PONE-D-20-21036

Skill Retention with Ultrasound Curricula

PLOS ONE

Dear Dr. Puri,

Thank you for submitting your manuscript to PLOS ONE. After careful consideration, we feel that it has merit but does not fully meet PLOS ONE’s publication criteria as it currently stands. Therefore, we invite you to submit a revised version of the manuscript that addresses the points raised during the review process.

We look forward to receiving your revised manuscript.

Kind regards,

Ezio Lanza, M.D.

Academic Editor

PLOS ONE

Journal Requirements:

Review Comments to the Author:

Reviewer #1: the authors have tried to investigate skill retention in ultrasound training; this is a laudable exercise, since not much evidence exists in this field; unfortunately, there is no information regarding the number of ultrasound scans that were performed by the control group. It is even possible that they have performed no scans after the training; detailed information about the number of scans in both groups is necessary information.

Reviewer #2: This paper presents randomized data of a 20 proctored lung ultrasound curriculum compared to the standard POCUS curriculum that is typical. I think the design is simple and sufficient for its intended goals. I think where to paper suffers is mostly placing this study in the broader context of IM curriculum POCUS studies that have been done. This is a minor but important point as I think properly compared to existing literature it will make this work more meaningful.

The statistical methods need to be better described (see below)

Line 71: Probably should include these citations

https://pubmed.ncbi.nlm.nih.gov/31125075/

https://www.ncbi.nlm.nih.gov/pmc/articles/PMC6638610/

Line 116: How does your assessment relate to this framework? It does seem like it encompasses the domains mentioned (Acquisition of knowledge, Application of obtained knowledge, Demonstration of technical competence, Integration into clinical practice, Re-certification – the 5th may not apply) but might explicitly state it does so.

https://pubmed.ncbi.nlm.nih.gov/30924088/

Box 1: Typo in anatomy #5 – “in”

Line 131: Methods: Would like the statistical considerations more clear.

For repeated measures ANOVA this was presumably to compare the same study participants results over time. And for the independent T tests this was mean tests scores between groups. Would make this more explicit. Was there any correction for multiple corrections?

Line 171: “Our competency based medical education study, used a learner” – no comma needed

Line 186-187: Needs citation and the relative % drop should be compared to other studies

Line 198: The assessment used was not validated as mentioned. This is unfortunate especially as validated tools do already exist. Besides the citation listed also see https://pubmed.ncbi.nlm.nih.gov/22124000/

Line 199: “Use of the same written assessment exam at each time point may have contributed to the improvement in test scores.” This needs to be further highlighted as a limitation and actually may invalidate the results over time have any bearing. I don’t see how the scores increased from immediately after the intervention to 6 and 12 mo. There is no statistical significance between groups so it does not change the interpretation of results but was a flaw in the design. (It is a well-established phenomenon on repeat testing i.e. MOCA).

Line 202. The conclusions are wanting. I think the key finding is that proctored scanning is the key to to observed noticeable difference in practical skill over time is important.

I think I would put it in context to this recent RCT (https://pubmed.ncbi.nlm.nih.gov/32118565/) that was surprising that machine access did not change retention. The major dogma has been access to trained faculty and machines have been the “barriers”. In this study the direct faculty involvement seemed to have a large impact that cannot be replaced by independent practice.

Reviewer #3: Thank you for the opportunity to review this manuscript. Ultrasound curriculum for internal medicine is an important topic, the authors describe the value of additional scanning for learning thoracic ultrasound. The manuscript is well written and clearly presented. However, the results are only described briefly and original data are lacking. Also there is a high risk of bias. I do have some major concerns.

Introduction

Lines 70-72 lack references

Methods

Line 79 How many residents were invited to participate?

Line 81 what are the inclusion criteria?

Line 82: how were they randomized?

Line 88: please describe the duration of the thoracic lecture and hands on scanning in the comprehensive course?

Line 99: please describe the protocol for the supervised ultrasound scans in relation the exams scan.

Line 101: when did the residents perform the scans? Which intervals? Or only immediately prior to the assessment?

Line 103: why were scans reviewed as they were directly supervised?

Line 105: proficiency of whom?

Line 109: did the control group performed any scans? How many? Who supervised them?

Line 113: were assessment dates known in advance? It is well known that students may practice more and study prior to an exam

Line 115-126: please add the exam questions and scoring system in the manuscript. The reader is unable to interpret the results without knowing the scoring system and range.

Line 115: was assessment of exams equally distributed between the 4 study investigators? For both groups? How did you manage to uniform assessment between assessors?

Results

Line 137: a baseline table is lacking with demographics and especially prior ultrasound experience

Table: it would be interesting to provide an additional table with the mean scores for every item, this will tell why the scores are different.

Please provide confidence intervals in tables. Have you tested for normal distribution of scores? Is mean applicable? There is a rather large SD.

Discussion

Line 173-175: I don’t understand the phrase “at their own pace”. If I understand correctly, both groups were able to perform ultrasound during the study period, but the intervention group made an additional 20 supervised scanning.

Line 184: did you compare scores within groups over time? Is the difference for the intervention group for 6 and 12 months significant?

Line 187: reference is lacking for statement

Line190: how did the difference in randomization influence your results as you mention this as a limitation?

Line 193: were exam scans equally distributed between study investigators? And for both groups?

Line 196: how many is some? In which groups? What part did they not complete?

Line 197: what do mean with external experts? And did they review the exams taken by the intervention group or only during development of the exams?

Line 199: was the practical exam also performed the same at each time point? And how did they compare to the additional 20 exams?

How many ultrasound did participants perform during the study period (beside the study protocol)? And for which applications?

---

## [Author Response · Author response to Decision Letter 0]

22 Sep 2020

Please see word filed uploaded.

Response to reviewers

1. Reviewer #1: the authors have tried to investigate skill retention in ultrasound training; this is a laudable exercise, since not much evidence exists in this field; unfortunately, there is no information regarding the number of ultrasound scans that were performed by the control group. It is even possible that they have performed no scans after the training; detailed information about the number of scans in both groups is necessary information.

Author’s response: We are grateful for your review and thoughtful critiques. The protocol did not include the number of scans completed by the control group, as it was not part of our standardized curricula. Both groups were able to perform scans outside of the study protocol. The protocol attempted to reflect usual training versus the intervention of 20 supervised scans. The house staff performs most of their diagnostic ultrasound training during their ICU rotations. Their exposure is dependent on patient population, faculty interest and their own interest. We included the information in the discussion and will consider these important points for future studies. 

Reviewer #2: This paper presents randomized data of a 20-proctored lung ultrasound curriculum compared to the standard POCUS curriculum that is typical. I think the design is simple and sufficient for its intended goals. I think where to paper suffers is mostly placing this study in the broader context of IM curriculum POCUS studies that have been done. This is a minor but important point as I think properly compared to existing literature it will make this work more meaningful.

Author’s response: Placing our study in the broader context of previous literature is important. We revised the introduction to reflect this point.

2. The statistical methods need to be better described (see below)

Author’s response: We have revised the methods to provide additional detail. 

3. Line 71: Probably should include these citations

https://pubmed.ncbi.nlm.nih.gov/31125075/ Mellor TE, Junga Z, Ordway S, et al. Not Just Hocus POCUS: Implementation of a Point of Care Ultrasound Curriculum for Internal Medicine Trainees at a Large Residency Program. Mil Med. 2019;184(11-12):901-906. doi:10.1093/milmed/usz124

https://www.ncbi.nlm.nih.gov/pmc/articles/PMC6638610/ LoPresti CM, Schnobrich DJ, Dversdal RK, Schembri F. A road map for point-of-care ultrasound training in internal medicine residency. Ultrasound J. 2019;11(1):10. Published 2019 May 9. doi:10.1186/s13089-019-0124-9

Author’s response: We have added the recommended references. 

4. Line 116: How does your assessment relate to this framework? It does seem like it encompasses the domains mentioned (Acquisition of knowledge, Application of obtained knowledge, Demonstration of technical competence, Integration into clinical practice, Re-certification – the 5th may not apply) but might explicitly state it does so.

https://pubmed.ncbi.nlm.nih.gov/30924088/ Kumar A, Kugler J, Jensen T. Evaluation of Trainee Competency with Point-of-Care Ultrasonography (POCUS): a Conceptual Framework and Review of Existing Assessments. J Gen Intern Med. 2019;34(6):1025-1031. doi:10.1007/s11606-019-04945-4

Author’s response: The conceptual framework and the Kumar et al reference was added to the manuscript. 

5. Box 1: Typo in anatomy #5 – “in”

Author’s response: We corrected the typo. 

6. Line 131: Methods: Would like the statistical considerations more clear.

Author’s response: We provided clarification of the statistical methods. 

7. For repeated measures ANOVA this was presumably to compare the same study participants results over time. And for the independent T tests this was mean tests scores between groups. Would make this more explicit. Was there any correction for multiple corrections? 

Author’s response: We have revised the statistical methods to provide clarity.

 Statistical Methods: 

We completed an analysis over time, examining the baseline, immediate posttest, at 6 months and 12 months. In order to determine which statistical test to run, we ran a Shapiro Wilke test and test of skewness, which indicated that some of the data were not normally distributed. In an effort to make the statistical testing consistent, we ran all repeated measures non-parametric testing using the Friedman test. An analysis of separate test scores performed via the Mann Whitney U test. Results are presented as median (interquartile range). We completed a Bonferroni correction in order to account for the 14 tests that were run. This created a significant level of p <=0.004. 

8. Line 171: “Our competency based medical education study, used a learner” – no comma needed

Author’s response: We made the correction. 

9. Line 186-187: Needs citation and the relative % drop should be compared to other studies

Author’s response: We had added a citation Kimura BJ, Sliman SM, Waalen J, Amundson SA, Shaw DJ. Retention of ultrasound skills and training in “point-of-care” cardiac ultrasound. Journal of the American Society of Echocardiography. 2016 Oct 1;29(10):992-7 for a study that evaluated knowledge retention of cardiac ultrasound at 1 year. Since we did not collect data when our subjects were out of the study, it is not exactly comparing “like” groups to each other, but it does quantify the amount of knowledge decay in medicine residents who undergo POCUS education. 

10. Line 198: The assessment used was not validated as mentioned. This is unfortunate especially as validated tools do already exist. Besides the citation listed also see https://pubmed.ncbi.nlm.nih.gov/22124000/

Author’s response: We appreciate the reviewer’s comments and suggested citations. We were unaware of the Bhaner et al. study prior to preparing the design for this study. The Bahner et al 2011 reference has been added to the manuscript. 

11. Line 199: “Use of the same written assessment exam at each time point may have contributed to the improvement in test scores.” This needs to be further highlighted as a limitation and actually may invalidate the results over time have any bearing. I don’t see how the scores increased from immediately after the intervention to 6 and 12 mo. There is no statistical significance between groups so it does not change the interpretation of results but was a flaw in the design. (It is a well-established phenomenon on repeat testing i.e. MOCA).

Author’s response: We included the limitation of using a repeated exam for this study. In our study, we were aiming to assess knowledge retention. We did not provide residents with the answers after the exam and all exams were numbered and returned to ensure no exams were missing.

12. Line 202. The conclusions are wanting. I think the key finding is that proctored scanning is the key to observed noticeable difference in practical skill over time is important.

Author’s response: We agree that based on the results of this study, proctored scanning resulted in an observed difference in practical skills over time. We have articulated this in the conclusion. 

13. I think I would put it in context to this recent RCT (https://pubmed.ncbi.nlm.nih.gov/32118565/) that was surprising that machine access did not change retention. The major dogma has been access to trained faculty and machines have been the “barriers”. In this study the direct faculty involvement seemed to have a large impact that cannot be replaced by independent practice.

Author’s response: We appreciate this feedback and added the 2020 citation - Kumar A, Weng Y, Wang L, et al. Portable Ultrasound Device Usage and Learning Outcomes Among Internal Medicine Trainees: A Parallel-Group Randomized Trial [published online ahead of print, 2020 Feb 11]. J Hosp Med. 2020;15(2):e1-e6. doi:10.12788/jhm.3351. The Kumar study provided IM physicians with a handheld ultrasound device with lectures over one year. This study did not include proctored scan acquisition with feedback. In our study of IM residents, we believe that a combination of lecture, hands on experience and observed image acquisition with real-time feedback provided a good framework for knowledge and skill retention. We included these points in the discussion. 

Reviewer #3: Thank you for the opportunity to review this manuscript. Ultrasound curriculum for internal medicine is an important topic, the authors describe the value of additional scanning for learning thoracic ultrasound. The manuscript is well written and clearly presented. However, the results are only described briefly and original data are lacking. Also there is a high risk of bias. I do have some major concerns.

Author’s response: We appreciate the reviewers comments. We have provided additional detail in the methods and results section of the paper. 

Introduction

14. Lines 70-72 lack references

Author’s response: Two references (Mellor et al and Lopresti et al) have been added to the introduction. 

Methods

15. Line 79 How many residents were invited to participate?

Author’s response: 36 residents were invited to participate in the study. We included this information in the results section of the manuscript. 

16. Line 81 what are the inclusion criteria?

Author’s response: Subjects were included if they were 1. a PGY1 or PGY2 resident , 2. provided informed consent and 3. attended the one-day ultrasound training course. We included this information in the manuscript. 

17. Line 82: how were they randomized?

Author’s response: There was a stratified randomization completed by PGY. Excel random numbers were used to determine which subject was assigned to which group. 

Author’s response:

18. Line 88: please describe the duration of the thoracic lecture and hands on scanning in the comprehensive course?

Author’s response: The methods have been modified to reflect the one- hour scanning for thoracic ultrasound and a one- hour thoracic lecture was provided. 

19. Line 99: please describe the protocol for the supervised ultrasound scans in relation the exams scan.

Author’s response: The manuscript has been revised to reflect that the resident completed a comprehensive thoracic and the study investigator provided feedback in real-time. Ten supervised scans were completed prior to the 6-month exam and another 10 prior to the 12-month exam. No predefined time interval was expected for exam completion, meaning that a subject could do more than one scan at a given time point to achieve their goal of 10 supervised scans in 6 months. 

20. Line 101: when did the residents perform the scans? Which intervals? Or only immediately prior to the assessment?

Author’s response: No predefined interval existed for when the scans were done, yet they were not done in bulk prior to the examination. Ten supervised scans were completed prior to the 6-month exam and another 10 prior to the 12-month exam.

21. Line 103: why were scans reviewed as they were directly supervised?

Author’s response: We believed that by directly supervising the exam, the subjects learned more from the dedicated investigator and real-time feedback improved skill retention. All of the investigators did not review the scans. We clarified this sentence in the manuscript.

22. Line 105: proficiency of whom?

Author’s response: The proficiency of the instructors. The manuscript has been revised to reflect the proficiency of the instructors. 

. Line 109: did the control group performed any scans? How many? Who supervised them?

Author’s response: We did not quantify the number of exams completed by the control group. A sentence has been included to indicate that both groups were able to perform scans unrelated to the study. Since we did not monitor scans performed by the control group, we are unable to comment on the number of scans or who if anyone supervised the subjects in the control group. 

23. Line 113: were assessment dates known in advance? It is well known that students may practice more and study prior to an exam

Author’s response: We appreciate these comments. We provided assessment dates in advance to assist with adherence to the study timelines. It is unknown if subjects practiced or studied prior to the reassessment. There were no limitations placed on students studying prior to an exam. In our opinion, additional studying would not be viewed negatively as this would be helpful to the residents.

24. Line 115-126: please add the exam questions and scoring system in the manuscript. The reader is unable to interpret the results without knowing the scoring system and range.

Author’s response: The exam questions and scoring system have been added to the manuscript as supplemental information. Of note, during the preparation of this table, we recognized that a written and practical score at 6 months in the control group and two practical scores at 12 months should not have been included in the analyses. We made the correction, reanalyzed all data and modified the manuscript as appropriate. 

25. Line 115: was assessment of exams equally distributed between the 4 study investigators? For both groups? How did you manage to uniform assessment between assessors?

Author’s response: Assessment of the scans was not equally distributed as the scans were completed on varying days and hours throughout the study period. Observation of scans was done based on availability of the investigator. The investigators agreed on a uniform assessment prior to study start. 

Results

26. Line 137: a baseline table is lacking with demographics and especially prior ultrasound experience.

Authors response: We have included sex of the participants to the PGY year, as no demographic data was captured from the subjects. We are unable to report on prior ultrasound experience, as we did not capture this information.

27. Table: it would be interesting to provide an additional table with the mean scores for every item, this will tell why the scores are different. 

Authors response: A table with the number of correct scores for each question has been added to the supplemental material. 

Please provide confidence intervals in tables. Have you tested for normal distribution of scores? Is mean applicable? There is a rather large SD. 

Author’s response: The tables have been revised to include 95% confidence intervals. An updated analysis examining medians (IQRs) instead of means is included. This change is reflected in the results section and within the tables.

Discussion

28. Line 173-175: I don’t understand the phrase “at their own pace”. If I understand correctly, both groups were able to perform ultrasound during the study period, but the intervention group made an additional 20 supervised scanning.

Author’s response: The manuscript has been modified to state that both groups were able to perform scans outside of the study requirements. We removed “at their own pace” and addressed how scans were performed prior to 6 months and 12 months in the methods section of the paper. 

29. Line 184: did you compare scores within groups over time? 

Author’s response: We compared the groups over time and data is available in figure 1.

30. Is the difference for the intervention group for 6 and 12 months significant?

Authors’s response – No. For the written exam, p = 0.08 and for the practical exam the p = 0.055. We added text to the manuscript “There were not significant differences in the 6 month nor 12 month scores in the assessment (written, p = 0.08; practical, p = 0.123)”. We have added this information to the text.

31. Line 187: reference is lacking for statement

Author’s response: A reference has been added along with context.

32. Line190: how did the difference in randomization influence your results as you mention this as a limitation?

Author’s response: The randomization scheme was developed for 36 potential participants stratified based on program year. Since only 28 of 36 residents provided consent, the control group and intervention were not equal in regards to PGY-1 and PGY-2 residents. We do not believe this influenced our results, yet we felt it was important to share the differences within the groups.

33. Line 193: were exam scans equally distributed between study investigators? And for both groups?

Author’s response: The scans were not equally distributed among study investigators. The manuscript has been updated to reflect this information.

34. Line 196: how many is some? In which groups? What part did they not complete?

Author’s response: One subject in the control group withdrew consent from the study and another was unable to complete the 6 and 12 month follow-up due to personal commitments. All subjects in the intervention group completed all exams required over the course of the year. The manuscript has been revised to reflect this information.

35. Line 197: what do mean with external experts? And did they review the exams taken by the intervention group or only during development of the exams?

Author’s response: Since we did not use validated questions, we used a variety of expert physicians to beta test the written and practical examination. The external experts were regional and or national leaders in the practice of POCUS. They only reviewed the development of the exams.

36. Line 199: was the practical exam also performed the same at each time point? And how did they compare to the additional 20 exams?

Author’s response: The practical exam was performed at the same time point as the written exam and it served as the basis for the 20 additional exams in the intervention group. 

37. How many ultrasound did participants perform during the study period (beside the study protocol)? And for which applications?

Author’s response: We did not track ultrasounds performed outside of the study protocol. Therefore, we are unable to provide information for this query.

---

## [Decision Letter · Decision Letter 1]

19 Oct 2020

PONE-D-20-21036R1

Skill Retention with Ultrasound Curricula

PLOS ONE

Dear Dr. Puri,

Thank you for submitting your manuscript to PLOS ONE. After careful consideration, we feel that it has merit but does not fully meet PLOS ONE’s publication criteria as it currently stands. Therefore, we invite you to submit a revised version of the manuscript that addresses the points raised during the review process.

We look forward to receiving your revised manuscript.

Kind regards,

Ezio Lanza, M.D.

Academic Editor

PLOS ONE

Reviewer #2: On re-review with the changes incorporated the Authors have made clear the design, findings and limitations and have better incorporated existing literature for comparison. My only comments now are stylistic:

Minor suggestions

Line 29, 74: ”valuable, but optimal…”

Line 32: from baseline instead of after baseline

Line 34: This sentence reads awkwardly

Line 80: How does this tie to skill retention

Line 194: I might say “An alternative yearlong lecture based model , even with personal HUD, showed no improvement in knowledge or image interpretation score.”

Reviewer #3: The authors have addressed my concerns. However two issues remain.

A major limitation is that it is unknown how many scans both groups performed during the study besides the study protcol. This study supports that proctored scanning helps, but the amount of the effect in unknown. If control group made no scans than 20 additonal scan is useful in increasing skills. If both groups made an additional 30 ultrasound, the most important explanation is that supervized scanning with bedside feedback is most important, not the amount.

it is still unclear to me how the practical assessment scores are computed. The supplement shows 23 points to be earned, but scores in tables are up to 86?

This needs to be explainedto be able to interpret the conclusions

---

## [Author Response · Author response to Decision Letter 1]

11 Nov 2020

Response to Reviewers

Reviewer #2: On re-review with the changes incorporated the Authors have made clear the design, findings and limitations and have better incorporated existing literature for comparison. My only comments now are stylistic:

Minor suggestions

Line 29, 74: ”valuable, but optimal…”

Line 32: from baseline instead of after baseline

Line 34: This sentence reads awkwardly

Line 80: How does this tie to skill retention

Line 194: I might say “An alternative yearlong lecture based model , even with personal HUD, showed no improvement in knowledge or image interpretation score.”

Authors’ response: Thank you for your kind remarks, detailed review of the revision and stylistic suggestions. We have incorporated all of the suggestions in the revised manuscript. 

Reviewer #3: The authors have addressed my concerns. However two issues remain.

A major limitation is that it is unknown how many scans both groups performed during the study besides the study protcol. This study supports that proctored scanning helps, but the amount of the effect in unknown. If control group made no scans than 20 additonal scan is useful in increasing skills. If both groups made an additional 30 ultrasound, the most important explanation is that supervized scanning with bedside feedback is most important, not the amount.

Authors’ response: We appreciate the reviewers concerns and have previously included this limitation in our previous manuscript. We did not consider tracking non-study related POCUS exams in either group as part of the protocol. Based on previous literature in this area, we also noted that Mellor et al, did not capture scans performed in non-study related activities and Kelm et al did not account for ultrasound exposure during the follow-up period, listing this as a limitation. However, we do appreciate the reviewers concerns and will consider this limitation in our future work in this area.

it is still unclear to me how the practical assessment scores are computed. The supplement shows 23 points to be earned, but scores in tables are up to 86?

This needs to be explainedto be able to interpret the conclusions

Authors’ response: The scores are reported as percentages, range 0-100% based on the number correct responses/the number of questions. We included a sentence in the abstract and the methods section of the paper to add clarification. Scores are reported as a percentage of the number of correct responses/number of questions (range 0 - 100%).

---

## [Editor Report · Decision Letter 2]

16 Nov 2020

Skill Retention with Ultrasound Curricula

PONE-D-20-21036R2

Dear Dr. Puri,

We’re pleased to inform you that your manuscript has been judged scientifically suitable for publication and will be formally accepted for publication once it meets all outstanding technical requirements.

Kind regards,

Ezio Lanza, M.D.

Academic Editor

PLOS ONE

---

## [Editor Report · Acceptance letter]

20 Nov 2020

PONE-D-20-21036R2 

Skill retention with ultrasound curricula 

Dear Dr. Puri:

I'm pleased to inform you that your manuscript has been deemed suitable for publication in PLOS ONE. Congratulations! Your manuscript is now with our production department. 

Kind regards, 

on behalf of

Dr. Ezio Lanza 

Academic Editor

PLOS ONE